End-point rapid detection of total and pathogenic Vibrio parahaemolyticus (tdh+ and/or trh1+ and/or trh2+) in raw seafood using a colorimetric loop-mediated isothermal amplification-xylenol orange technique

Lamalee Aekarin 1
Saiyudthong Soithong 2
Changsen Chartchai 1
Kiatpathomchai Wansika 3
Limthongkul Jitra 1
Naparswad Chanita 1
Sukphattanaudomchoke Charanyarut 1
Chaopreecha Jarinya 1
Senapin Saengchan 4 5
Jaroenram Wansadaj kungbtram@gmail.com 3
Buates Sureemas sbuates@hotmail.com 1
1 Department of Microbiology, Faculty of Science, Mahidol University , Bangkok , Thailand
2 Institute of Food Research and Product Development, Kasetsart University , Bangkok , Thailand
3 Bioengineering and Sensing Technology Research Team, National Center for Genetic Engineering and Biotechnology (BIOTEC), National Science and Technology Development Agency (NSTDA) , Pathum Thani , Thailand
4 Center of Excellence for Shrimp Molecular Biology and Biotechnology, Faculty of Science, Mahidol University , Bangkok , Thailand
5 National Center for Genetic Engineering and Biotechnology (BIOTEC), National Science and Technology Development Agency (NSTDA) , Pathum Thani , Thailand
Beddoe Travis
Electronic publication date: 2024 Jan 3
Publication date: 2024
Volume: 12
Electronic Location ID: e16422
Received 2023 May 22; Accepted 2023 Oct 17
Copyright: ©2024 Lamalee et al.
Copyright year: 2024
Copyright holder: Lamalee et al.
License: This is an open access article distributed under the terms of the Creative Commons Attribution License, which permits unrestricted use, distribution, reproduction and adaptation in any medium and for any purpose provided that it is properly attributed. For attribution, the original author(s), title, publication source (PeerJ) and either DOI or URL of the article must be cited.
License URL: https://creativecommons.org/licenses/by/4.0/

Keywords: Loop-mediated isothermal amplification-xylenol orange (LAMP-XO), V. parahaemolyticus, Seafood, Rapid detection, Colorimetric technique

Funding: Kasetsart University Research and Development Institute National Research Council of Thailand This research project was supported by grants from the Kasetsart University Research and Development Institute and the National Research Council of Thailand (NRCT). The funders had no role in study design, data collection and analysis, decision to publish, or preparation of the manuscript.

==============================
Background

Vibrio parahaemolyticus is the leading cause of bacterial seafood-borne gastroenteritis in humans worldwide. To ensure seafood safety and to minimize the occurrence of seafood-borne diseases, early detection of total V. parahaemolyticus (pathogenic and non-pathogenic strains) and pathogenic V. parahaemolyticus (tdh+ and/or trh1+ and/or trh2+) is required. This study further improved a loop-mediated isothermal amplification (LAMP) assay using xylenol orange (XO), a pH sensitive dye, to transform conventional LAMP into a one-step colorimetric assay giving visible results to the naked eye. LAMP-XO targeted rpoD for species specificity and tdh, trh1, and trh2 for pathogenic strains. Multiple hybrid inner primers (MHP) of LAMP primers for rpoD detection to complement the main primer set previously reported were designed by our group to maximize sensitivity and speed.

Methods

Following the standard LAMP protocol, LAMP reaction temperature for rpoD, tdh, trh1, and trh2 detection was first determined using a turbidimeter. The acquired optimal temperature was subjected to optimize six parameters including dNTP mix, betaine, MgSO4, Bst 2.0 WarmStart DNA polymerase, reaction time and XO dye. The last parameter was done using a heat block. The color change of the LAMP-XO result from purple (negative) to yellow (positive) was monitored visually. The detection limits (DLs) of LAMP-XO using a 10-fold serial dilution of gDNA and spiked seafood samples were determined and compared with standard LAMP, PCR, and quantitative PCR (qPCR) assays. Subsequently, the LAMP-XO assay was validated with 102 raw seafood samples and the results were compared with PCR and qPCR assays.

Results

Under optimal conditions (65 °C for 75 min), rpoD-LAMP-XO and tdh-LAMP-XO showed detection sensitivity at 102 copies of gDNA/reaction, or 10 folds greater than trh1-LAMP-XO and trh2-LAMP-XO. This level of sensitivity was similar to that of standard LAMP, comparable to that of the gold standard qPCR, and 10-100 times higher than that of PCR. In spiked samples, rpoD-LAMP-XO, tdh-LAMP-XO, and trh2-LAMP-XO could detect V. parahaemolyticus at 1 CFU/2.5 g spiked shrimp. Of 102 seafood samples, LAMP-XO was significantly more sensitive than PCR (P < 0.05) for tdh and trh2 detection and not significantly different from qPCR for all genes determined. The reliability of tdh-LAMP-XO and trh2-LAMP-XO to detect pathogenic V. parahaemolyticus was at 94.4% and 100%, respectively.

Conclusions

To detect total and pathogenic V. parahaemolyticus, at least rpoD-LAMP-XO and trh2-LAMP-XO should be used, as both showed 100% sensitivity, specificity, and accuracy. With short turnaround time, ease, and reliability, LAMP-XO serves as a better alternative to PCR and qPCR for routine detection of V. parahaemolyticus in seafood. The concept of using a one-step LAMP-XO and MHP-LAMP to enhance efficiency of diagnostic performance of LAMP-based assays can be generally applied for detecting any gene of interest.

Introduction

Seafood products are widely consumed globally and play an important role in the economic market. Even though seafood consumption is suggested as a part of a healthy diet (Hosomi, Yoshida & Fukunaga, 2012), several pathogens related with adverse human health effects including gastrointestinal diseases are present in seafood (Choudhury et al., 2022). Because of the growth in global consumption of seafood products, inspection of seafood quality is the primary way to inhibit contamination from seafood-borne pathogens (FAO, 2020; Choudhury et al., 2022). Among seafood-borne pathogens, Vibrio parahaemolyticus, a Gram-negative, rod-shaped bacterium, is a leading cause of seafood-borne bacterial gastroenteritis in humans. Global gastroenteritis caused by V. parahaemolyticus is normally due to consumption of raw or undercooked contaminated seafood, especially shellfish (Raszl et al., 2016; Centers for Disease Control and Prevention , 2019). Vibrio parahaemolyticus is naturally present in marine environments globally. It is often associated with aquatic products as well as shellfish, shrimp, and fish (Letchumanan et al., 2019; Changsen et al., 2023). This bacterium accounts for a considerable increase of seafood-borne infections worldwide (Letchumanan, Chan & Lee, 2014).

Vibrio parahaemolyticus possesses different virulence factors. The major ones are thermostable direct hemolysin (TDH) encoded by the tdh gene and TDH-related hemolysins (TRH) encoded by the trh gene, both of which are present mostly in clinical strains (Raghunath, 2015; Cai & Zhang, 2018). TDH is a heat-resistant, pore-forming toxin comprising of 156 amino acids (Li et al., 2019). It forms pores on erythrocyte membrane, allowing water and ions to flow through the membrane leading to erythrocyte lysis. It also exerts several mainly biological activities including cytotoxicity, cardiotoxicity, and enterotoxicity (Cai & Zhang, 2018). TRH is a heat-labile toxin composing of 189 amino acids. It has similar biological activities to the TDH (Honda, Ni & Miwatani, 1988; Nishibuchi et al., 1989). The trh and tdh genes share 54.8–68.8% identity in their sequences (Kishishita et al., 1992). The trh gene can be subdivided into trh1 and trh2 which share 84% homology (Nishibuchi et al., 1989; Kishishita et al., 1992). It is known that all pathogenic V. parahaemolyticus strains harbor tdh and/or trh genes while the non-pathogenic strains lack both tdh and trh genes (Honda & Iida, 1993; Nishibuchi & Kaper, 1995). Therefore, tdh and trh genes are considered as molecular markers for V. parahaemolyticus pathogenicity.

Many V. parahaemolyticus infections are epidemiologically related with seafood consumption, especially shellfish contaminated with pathogenic strains (Raszl et al., 2016). Accordingly, controlling V. parahaemolyticus contamination in seafood can effectively prevent seafood-borne diseases. Moreover, ecological and epidemiological surveillance on the prevalence of pathogenic V. parahaemolyticus strains is needed since virulence genes can be horizontally transferred to non-virulent strains (Waldor & Mekalanos, 1996). Several methods are available for detection of tdh and trh genes of V. parahaemolyticus including polymerase chain reaction (PCR), quantitative PCR (qPCR), and droplet digital PCR. However, these advanced methods are time-consuming and need expensive apparatus, expensive reagents as well as trained personnel (Xu et al., 2018; Lei et al., 2020; Guan et al., 2021). Therefore, developing a simple diagnostic method with high sensitivity and specificity would be essential.

Loop-mediated isothermal amplification (LAMP), a DNA-based amplification assay that amplifies nucleic acids under a single temperature is one such tool, having the potential to be a DNA-based point-of-care (POC) diagnostic method (Notomi et al., 2000). Unlike PCR and qPCR, LAMP assay can be done in a heat block or water bath. Recently, many LAMP assays have been employed to detect tdh and trh genes of V. parahaemolyticus (Yingkajorn et al., 2014; Yan et al., 2017; Anupama et al., 2021). In general, the detection of LAMP products/amplicons can be done via (1) a direct observation of the white precipitate (turbidity) of magnesium pyrophosphate, Mg2P2O7, (a by-product of LAMP reaction) and (2) indirect methods including, turbidity measurement by a turbidimeter (EIKEN Co. Ltd.), fluorescent observation under UV light, agarose gel electrophoresis (AGE), and lateral flow dipstick (LFD) detection (Mori et al., 2001; Prompamorn et al., 2011; Wong et al., 2018). However, the direct method monitoring with the naked eye can give ambiguous readout, especially in weakly positive samples. Although this weakness can be avoided by using a turbidimeter or a UV light transilluminator, such devices are costly and required additional steps to work on (Wong et al., 2018). Likewise, AGE and LFD require additional costs, post-amplification steps and opening of a reaction tube, thereby risking possible cross-contamination. To overcome these drawbacks and to improve the overall diagnostic performance of direct LAMP assay readout, a simpler colorimetric method was explored. In 2019, Jaroenram, Cecere & Pompa (2019) exploited xylenol orange (XO) to detect Escherichia coli DNA. The XO is a low-cost pH indicator whose color changes from violet to yellow at pH < 6.7, allowing a chance to detect the progress of LAMP reaction directly via the naked eye. To illustrate, during LAMP amplification, large amounts of Mg2P2O7 and protons (H+) are generated, resulting in a significant pH drop from initial alkaline pH values (8.5–9.0) to a final acidic pH value of approximately 6.0–6.5 when LAMP reaction is performed in low concentrations of buffer or non-buffered solution (Tanner, Zhang & Evans Jr, 2015). In the presence of XO, the presence of target DNA in test samples will trigger the change of the reaction hue from purple to yellow (positive readout/LAMP amplicon buildup). A lack of detection targets will present the original violet hue of the reaction (negative result). The result can be seen easily by the naked eye. To the best of our knowledge, LAMP-XO strategy has not been applied to detect Vibrio spp. Herein, we have developed LAMP-XO assay, and validated whether it would rapidly, sensitively, and accurately detect total V. parahaemolyticus (pathogenic and non-pathogenic strains) and pathogenic V. parahaemolyticus (tdh+ and/or trh1+ and/or trh2+) in raw seafood. Based on 102 raw seafood samples, the assay is as efficient as the existing molecular assays but cheaper, faster, and easier to use, thus more suitable for point of care (POC) testing.

Materials & Methods

Bacterial strains, culture conditions, and genomic DNA template preparation

Vibrio parahaemolyticus reference stains, DMST 15285 (tdh+/trh−) (obtained from the Department of Medical Science, Ministry of Public Health, Nonthaburi Province, Thailand), TISTR 1596 or ATCC 17802 (tdh−/trh2+) (obtained from the Thailand Institute of Scientific and Technological Research), and Vp10/5 (tdh+/trh1+) (obtained from an oyster, Chatuchak District, Bangkok, Thailand in 2018) were utilized as the positive controls to optimize and validate LAMP-XO assay. The reference stains were cultured in 5 mL tryptic soy broth (TSB) (Merch KGaA, Darmstadt, Germany) supplemented with 2% NaCl (BDH Poole, USA) in a shaking incubator at 250 rpm for 16 h at 37 °C. A volume of three mL of the culture broth was transferred into a 1.5-mL microcentrifuge tube and centrifuged at 10,000 rpm for 2 min to remove the supernatant. The pellet was resuspended in Tris-EDTA (TE) buffer (Invitrogen, Grand Island, NY, USA), and subjected to genomic DNA (gDNA) extraction using GenUP™ kit (Biotechrabbit, Berlin, Germany) according to the manufacturer’s protocol. The gDNA concentration was measured using a nanodrop (DS-11 FX+, spectrophotometer/fluorometer; DeNovix, Wilmington, DE, USA). The gDNA solution was aliquoted and kept at −80 °C.

Sample collection, V. parahaemolyticus detection, storage, recovery, and gDNA template preparation

A total of 102 raw seafood samples were used in this study. Of the 102 samples, 83 were purchased from fresh markets and supermarkets in Bangkok, Thailand of which 30 and 53 were purchased in 2018 and 2021 to 2022, respectively. Of the 102 samples, 16 samples were V. parahaemolyticus isolates obtained from Pacific white shrimp collected from different shrimp farms in eastern Thailand in 2013 and three samples were V. parahaemolyticus isolates obtained from Pacific white shrimp collected from North Vietnam in 2016. Eighty three samples purchased from markets comprised of six crabs (five blue swimming crabs and one red swimming crab), 15 fish (five groupers, five giant sea perches, two mackerels, one ornate threadfin bream, one bluefin tuna, and one salmon), 31 mollusc shellfish (15 oysters, six green mussels, five blood cockles, two short-necked clams, two spiral babylon snails, and one scallop), 21 shrimp (one giant tiger prawn and 20 Pacific white shrimp) and 10 squids (six splendid squids, three octopuses, and one giant squid tentacle). For V. parahaemolyticus detection of the purchased seafood, the samples were separately put into a sterile plastic bag and taken to a laboratory in a cooler bag containing ice and were handled within 2 h after sample purchasing. Briefly, 2.5 g of each sample was cut aseptically and immersed into 22.5 mL of TSB (Merch KGaA, Germany) supplemented with 2% NaCl (BDH Poole, Rahway, NJ, USA) for bacterial enrichment. The sample was gently mixed by hand and incubated at room temperature (RT) for 30 min before being removed from culture broth which was further incubated at 37 °C for 16 h. A total of three mL of culture broth was subjected to gDNA isolation. The extracted gDNA was used as a template for LAMP-XO, standard LAMP, qPCR, and conventional PCR assays. For V. parahaemolyticus isolation, one loop full of the culture broth was streaked onto thiosulfate-citrate-bile salts-sucrose agar (TCBSA) (BD, Sparks, USA) on which V. parahaemolyticus produced opaque and blue-green color with 2–3 mm in diameter colonies. The presumptive V. parahaemolyticus colonies were picked up separately and transferred to CHROMagar™ Vibrio (CHROMagar, Paris, France) on which V. parahaemolyticus produced mauve color colonies. For long-term storage, a single mauve colony of V. parahaemolyticus on CHROMagat™ Vibrio was cultured in TSB (Merch KGaA, Germany) supplemented with 2% (w/v) NaCl (BDH Poole, Rahway, NJ, USA) at 37 °C for overnight. A total of 400 µL of 50% glycerol (Sigma-Aldrich, St.Louis, MO, USA) and 600 µL of an overnight culture were mixed aseptically in a sterile 1.5-mL microcentrifuge tube and kept at −80 °C. For bacterial recovery, ice crystals on top of a glycerol stock were aseptically scraped and transferred into TSB supplemented with 2% NaCl and incubated in a shaker incubator at 37 °C with 250 rpm shaking for overnight. The presumptive culture was subjected to species-specific rpoD-qPCR assay for confirmation of species.

Optimization and validation of LAMP-xylenol orange assay

Optimization of LAMP-xylenol orange (LAMP-XO) assay was carried out using 4 sets of previously described primers targeting four different genes: Set 1 for rpoD gene for total V. parahaemolyticus (pathogenic and non-pathogenic strains) detection (Nemoto et al., 2011; Lamalee et al., 2023) and sets 2–4 for tdh, trh1, and trh2 genes, respectively, for V. parahaemolyticus pathogenic strain detection (Table 1) (Nemoto et al., 2009; Yamazaki et al., 2010). The optimization for each gene was done separately. For rpoD detection, in addition to the main primers utilized, we designed four additional primers (loop forward and loop backward 2; F1c2 and B1c2, and forward inner and backward inner 2; FIP2 and BIP2) (Table 1) to further improve the reaction kinetics (Jaroenram et al., 2022). The primers were examined for possible cross dimerization by basic local alignment search tool (BLAST) (https://blast.ncbi.nlm.nih.gov/Blast.cgi).

Table 1 LAMP primers and conditions used in this study.

Primer name	Sequence (5′ to 3′)	Target gene	Concentration ( pmol/µL )	Reference	
rpoD-FIP	TGAATACGTCTAGCATCATTTCGTCGATCAATGAGTACGGCTACGA	rpoD	20	Nemoto et al. (2011)	
rpoD-BIP	ACAGCAATGGATCGCGTTCCGATTTCTTCGGCATTTTGCC		20		
rpoD-F3	ACCAGCTACGCAGCACA		20		
rpoD-B3	CACTTGATTCGTTACCAGTGAATAGG		20		
rpoD- LF	GCAACGGTTGCTTTCGG		20		
rpoD- LB	GTTTGATCATGAAGTCTGTGG		20		
Extra primer				Lamalee et al. (2023)	
rpoD-FIB2	TGGTGTTAGACGGAATTCTTTTCGATCAATGAGTACGGCTACGA		20		
rpoD-BIP2	CACCTAGTGAACGAACTTCTTTTCGATTTCTTCCCCATTTTGCC		20		
rpoD-F1c2	TGGTGTTAGACGGAATTC		20		
rpoD-B1c2	CACCTAGTGAACGAACTTC		20		
tdh-FIP	CTTATAGCCAGACACCGCTGCGGTTGACATCCTACATGACTGTG	tdh	40	Nemoto et al. (2009)	
tdh-BIP	CGGTCATTCTGCTGTGTTCGTTCTTCACCAACAAAGTTAGCTACAG		40		
tdh-F3	GTCTCTGACTTTTGGACAAACCG		10		
tdh-B3	CTACATTAACAAAATATTCTGGAGTTTCATCC		10		
tdh-LF	CCGCTGCCATTGTATAGTCTTT		40		
tdh-LB	CAGATCAAGTACAACTTCAACATTCCT		40		
trh1-FIP	AGGCTTGTTTTTTCTGATTTTGTGACTACACAATGGCTGCTCT	trh1	40	Yamazaki et al. (2010)	
trh1-BIP	TCTTCTGTTAGTGATTTCGTTGGTTTTCATCCAAATACGTTACACT		40		
trh1-F3	GCGCCTATATGACGGTAA		5		
trh1-B3	ACATTGACGAAATATTCTGGC		5		
trh1-LF	AGACCGTTGARAGGCC		20		
trh2- FIP	CCGATTGACCGTATACATCTTTGTTGTGGAGGACTATTGGACAA	trh2	40	Yamazaki et al. (2010)	
trh2- BIP	TCAAAGTGGTTAAGCGCCTATATGCCATSTTTATAACCAGAAAGAGC		40		
trh2- F3	CATCAATACCTTTTCCTTCTCC		5		
trh2- B3	GCTTGTTTTCTCTGATTTTGTG		5		
trh2- LF	TGGTTTTCTTTTTATGKTTCGGT		20		
trh2- LB	ATGGTCAYAACTATACRATGGC		20		

Briefly, the protocol (Table S1) was done in a 25-µL reaction mixture containing each target-specific primer set (forward inner primer (FIP), backward inner primer (BIP), forward outer primer (F3), backward outer primer B3), forward loop primer (LF), and backward loop primer (LB)) at different amounts (Table 1), dNTP mix (New England Biolabs, Ipswich, MA, USA), betaine (Sigma-Aldrich, St.Louis, MO, USA) MgSO4 (New England Biolabs, Ipswich, MA, USA), Bst 2.0 WarmStart DNA polymerase (New England Biolabs, Ipswich, MA, USA), 2.5 µL of 1× low-buffer solution with pH 8.5 (100 mM (NH4)2SO4, 500 mM KCl, 20 mM MgSO4, and 1% Tween-20), and 1 µL of a gDNA template. The final volume was adjusted to 25 µL using UltraPure™ distilled water (DW) (Invitrogen, Grand Island, Germany). The negative control containing only DW (no gDNA templates) was included in each run. LAMP reaction was done in a Loopamp Realtime Turbidimeter LA-320C (Eiken Chemical Co Ltd, Tokyo, Japan) at a given condition (temperature and time) followed by DNA polymerase inactivation at 80 °C for 5 min. Following this initial protocol, the optimal incubation temperature was first determined, in that the LAMP reactions were carried out at various temperatures (60, 63, and 65 °C) for 75 min. The obtained optimal temperature was then subjected to optimizing six respective parameters: dNTP mix (1.2–1.8 mM), betaine (0.2–0.8 M), MgSO4 (4–10 mM), Bst 2.0 WarmStart DNA polymerase (6–12 U), reaction time (30, 45, 60, and 75 min), and XO dye, (0.03–0.12 mM) (Table S2). The last parameter was performed in a heat block, and the result was inspected visually. Color of LAMP-XO was changed from purple to yellow in a positive test while color was still purple in a negative test. To confirm LAMP-XO results, LAMP amplicons were analyzed by 3% agarose gel electrophoresis (AGE) (Vivantis, Malaysia), stained with ethidium bromide (Invitrogen, Waltham, MA, USA) and visualized under UV illumination. Each parameter used 1 µL gDNA of 106 copies/µL/reaction as a template except for the incubation temperature and time that used 1 µL aliquot of 10-fold serially diluted DNA (104, 103, 102, 10 copies/µL) instead. The DNA copy has been calculated using the formular “amount of DNA (ng) × 6.022 × 1023/ length of a DNA template (bp) × 1  × 109 × 650” (https://www.technologynetworks.com/tn/tools/copynumbercalculator). For each primer set/target gene, any given temperature, time, and components’ concentration that maximize DNA amplification based on signal intensities by the turbidimeter and the degree of color change from purple (negative) to yellow (postitive) was selected to establish the standard LAMP-XO protocol.

Standard LAMP assay

The standard LAMP assay was non-colorimetric, adopted from the standard LAMP protocol suggested by New England Company Ltd. (https://international.neb.com/protocols/2014/06/17/loop-mediated-isothermal-amplification-lamp). It was used as a control assay to test the efficiency of LAMP-XO. Its reaction components were almost similar to those of the optimized LAMP-XO, only excluding XO, and the low-buffer that was substituted by 1× ThermoPol-supplied reaction buffer (20 mM Tris-HCl, 10 mM (NH4)2SO4, 10 mM KCl, 2 mM MgSO4 and 0.1% Triton X-100, pH 8.8). The LAMP reaction was performed in a Loopamp Realtime Turbidimeter LA-320C (Eiken Chemical Co Ltd, Tokyo, Japan), at 65 °C (75 min for rpoD, tdh, trh1, and trh2). The results were reported in a real-time amplification plot format. For result confirmation, LAMP products were inspected using 3% AGE.

Conventional PCR assay

Conventional PCR assay was performed using species-specific toxR primers for V. parahaemolyticus detection and tdh, trh1, and trh2 primers for V. parahaemolyticus pathogenic strain detection (Table S3) (Tada et al., 1992; Kim et al., 1999; Messelhäusser et al., 2010). PCR reaction was carried out in a 25-µL reaction mixture containing 12.5 µL GoTaq® Green Master Mix solution (Promega, Madison, USA), 0.4 µM each forward primer and backward primer for toxR, tdh, trh1, and trh2 primers and 1 µL of a gDNA template. A final volume of a 25-µL reaction mixture was adjusted with UltraPure™ DW (Invitrogen, Grand Island, Germany). The PCR amplification was done in T100 Thermal Cycler No. 186-1096 (Bio-Rad, Hercules, CA). PCR products were analyzed using 1.5% AGE (Vivantis, Malaysia), stained with ethidium bromide (Sigma-Aldrich, St.Louis, MO, USA) and observed under a UV light. A positive control and a negative control were included in each run.

Quantitative PCR assay

A quantitative PCR (q-PCR) assay was done using species-specific rpoD primers for V. parahaemolyticus detection and tdh, trh1, and trh2 primers for V. parahaemolyticus pathogenic strain detection (Table S4) (Nemoto et al., 2009; Messelhäusser et al., 2010; Yamazaki et al., 2010; Nemoto et al., 2011). A 20-µL reaction mixture consisted of 10 µL of 2× KAPA SYBR FAST qPCR Master Mix Universal (Sigma-Aldrich, St.Louis, MO, USA), 10 mM each forward primer and backward primer, and 1 µL aliquot of 20 ng/µL of a gDNA template. A final volume was adjusted using UltraPure™ DW (Invitrogen, Grand Island, Germany). The qPCR amplification was carried out in a Rotor-Gene Q R10116106 (Qiagen Hilden, Germany). A positive control, a negative control, and a standard curve were included in each run.

Comparative detection limit of LAMP-XO, standard LAMP, conventional PCR, and qPCR assays by gDNA

The concentrations of gDNA from the reference strains, DMST 15285, TISTR 1596 or ATCC 17802, and Vp10/5, were measured using a nanodrop (DS-11 FX+, spectrophotometer/fluorometer; DeNovix, Wilmington, DE, USA) and subjected to serial dilution in a 10-fold manner from 104–100 copies/µL. The DNA solution was used as a template for LAMP-XO, standard LAMP, qPCR, and conventional PCR assays.

Comparative detection limit of LAMP-XO, standard LAMP, conventional PCR, and qPCR assays by spiked seafood samples

Three Pacific white shrimp from a supermarket were sterilized by autoclaving. Lack of V. parahaemolyticus contamination was later confirmed by a culture method using TCBSA (Difco Laboratories, Sparks, MD, USA). Briefly, 2.5 g of each Pacific white shrimp was spiked with one mL aliquot of a 10-fold serial dilution of overnight culture of V. parahaemolyticus reference strains to generate inoculating levels of 104–100 colony-forming unit (CFU)/2.5 g sample. The inoculating levels were calculated based on the assumption of OD600 = 1 = 8  × 108 CFU/mL. A negative control was a non-spiked sample. Each sample was adjusted to 22.5 mL using TSB supplemented with 2% NaCl, mixed gently by hands and stored at RT for 30 min. All samples were further incubated at 37 °C for 4 h prior to gDNA extraction. The obtained gDNA was subjected to LAMP-XO, qPCR, and conventional PCR assays.

Detection of total V. parahaemolyticus(pathogenic and non-pathogenic strains) and pathogenicV. parahaemolyticus (tdh+and/ortrh1+and/ortrh2+) in raw seafood samples by LAMP-XO andstatistical analysis

For LAMP-XO validation using seafood samples, LAMP-XO reaction for each target gene was carried out using optimal reagent concentrations and conditions as shown in Table 2. Final results of LAMP-XO validation were cross-compared with conventional PCR and qPCR. Specimens were classified as true positive, true negative, false positive, or false negative for each test under evaluation compared to qPCR as a gold standard. The clinical sensitivity, specificity, positive predictive values (PPV), negative predictive values (NPV), diagnostic accuracy, percent overall agreement (POA), and 95% CIs of LAMP-XO, conventional PCR, and qPCR assays were analyzed by SAS software (SAS Institute Inc., Cary, NC, USA). The clinical sensitivity was calculated as (number of true positives)/(number of true positives + number of false negatives) × 100, and the clinical specificity was calculated as (number of true negatives)/(number of true negatives + number of false positives) × 100. The PPV was calculated as (number of true positives)/(number of true positives + number of false positives) × 100, and the NPV was calculated as (number of true negatives)/(number of true negatives + number of false negatives) × 100. The accuracy was calculated as (number of true positives + number of true negatives)/(total number of patients) × 100. The degree of agreement between two diagnostic tests was measured by the concordance response rate (percentage of responses with both positive or both negative results). The POA indicates the percentage of relationship between varied results of comparative and reference methods. Note that the clinical (statistical) sensitivity refers to the ability of the validated assay to correctly identify real positive samples (V. Parahaemolyticus contaminated samples), while the clinical (statistical) specificity refers to the ability of the validated assay to correctly identify real negative samples (V. Parahaemolyticus-free samples). The significant difference between two detection methods, i.e., LAMP-XO and qPCR and LAMP-XO and conventional PCR for each gene was analyzed with a McNemar chi-square test and a P value < 0.05 was considered significant.

Table 2 LAMP-XO conditions, primers, and reagents’ concentrations.

Target gene	Optimal parameter of LAMP-XO reaction in a final volume of 25 µL	
	Temperature
(° C)	dNTP mix
(mM)	MgSO 4
(mM)	Betaine
(M)	Bst 2.0 WS DNA polymerase (U)	Reaction time (min)	XO
(mM)	Low buffer	
rpoD	65	1.6	8	0.8	8	75	0.06	1 ×	
tdh	65	1.4	8	0.8	8	75	0.06	1 ×	
trh1	65	1.2	8	0.8	8	75	0.03	1 ×	
trh2	65	1.2	8	0.8	8	75	0.03	1 ×	
Notes.

The concentrations of primers for each target gene are shown in Table 1.

Results

Optimization of LAMP-XO assay conditions

LAMP reactions for each target gene (rpoD, tdh, trh1, and trh2) (Table 1) were performed in a low-buffer solution (pH 8.5) based on the method previously reported to leverage the effect of a pH drop for colorimetric product (Jaroenram, Cecere & Pompa, 2019). Since LAMP reaction standard temperatures range from 60–65 °C, the incubation temperature at which a template could be best amplified was firstly optimized, followed by LAMP reagents, reaction time, and XO concentrations (Table S2).

In the LAMP reaction cocktail, the optimal XO concentration would qualitatively generate the most substantial colorimetric change from purple (negative) to yellow (positive) upon a decrease in pH (< 6.7) in the presence of the LAMP reaction by-products. The XO concentration producing the most distinctive color difference between the positive and negative test results was selected (Table S5). Using a high XO concentration is a concern since it interferes in a weak positive reaction. Overall, different target genes have different components’ concentrations, except the incubation temperature (65 °C) and amplification time (75 min) which is in common as reported in Table 2. The performance of optimal LAMP-XO was compared with conventional PCR and qPCR assays (Table S6).

Comparative detection limit of LAMP-XO, standard LAMP, conventional PCR, and qPCR assays using a 10-fold serial dilution of gDNA

To test the analytical sensitivity of the assays, LAMP-XO, standard LAMP, PCR, and qPCR assays were conducted for each target gene/primer set with the same set of a gDNA template ranging from 104–100 copies/reaction (Table 3, Fig. 1). Primers targeting rpoD were used for total V. parahaemolyticus (pathogenic and non-pathogenic strains) detection in LAMP-XO, standard LAMP, and qPCR assays while primers targeting toxR were used for total V. parahaemolyticus (pathogenic and non-pathogenic strains) detection in PCR. Primers targeting tdh, trh1, and trh2 were used for pathogenic V. parahaemolyticus (tdh+ and/or trh1+ and/or trh2+) detection. For LAMP-XO assay, rpoD and tdh primers indicated positive amplification at 104–102 copies, while trh1 and trh2 primers did at 104–103 copies. Thus, the DLs of the naked eye detection of LAMP-XO for rpoD and tdh genes were 102copies/reaction and for trh1 and trh2 genes were 103 copies/reaction (Fig. 1A, top). These colorimetric results were in agreement with those confirmed by AGE (Fig. 1A, bottom). Standard LAMP assay determined DLs of 102 copies/reaction for rpoD and tdh and 103 copies/reaction for trh1 and trh2 for both turbidity measurement (Fig. 1B, top) and AGE (Fig. 1B, bottom). PCR produced DLs at 103 copies for toxR and 104 copies for tdh, trh1, and trh2 (Fig. 1C), while qPCR assay produced DLs at 101 copies/reaction for tdh and 102 copies/reaction for rpoD, trh1, and trh2 (Fig. 1D).

Table 3 Comparative detection limit (DL) of LAMP-XO, standard LAMP, conventional PCR, and qPCR assays using a 10-fold serial dilution of gDNA (104–100 copies/reaction).

Primer	10-fold serial dilution of gDNA (10 4 - 10 0 copies/reaction)	
	LAMP-XO	Standard LAMP	Conventional PCR	qPCR	
toxR/rpoD	102	102	103	102	
tdh	102	102	104	101	
trh1	103	103	104	102	
trh2	103	103	104	102	

Figure 1 Comparative sensitivity of LAMP-XO, standard LAMP, conventional PCR, and qPCR assays for the detection of total V. parahaemolyticus and pathogenic V. parahaemolyticus using 104–100 copies of gDNA/reaction.

The rpoD primer was used for total V. parahaemolyticus detection in LAMP-XO, standard LAMP, and qPCR assays while the toxR primer was used in PCR assay. The tdh, trh1, and trh2 primers were used for pathogenic V. parahaemolyticus detection in all assays. (A) LAMP-XO assay detected by the naked eye (top) and AGE (bottom); (B) standard LAMP assay detected by a turbidimeter (top) and AGE (bottom); (C) conventional PCR assay detected by AGE; (D) Quantitative PCR (q-PCR) assay. (A), (B), and (D) L1: 104 copies/reaction, L2: 103 copies/reaction, L3: 102 copies/reaction, L4: 101 copies/reaction, L5: 100 copies/reaction; (C) L1-L3: 104 copies/reaction, L4-L6: 103 copies/reaction, L7-L9: 102 copies/ reaction; M: 1 kb DNA ladder, N: Negative control.

The LAMP-XO assay exhibited comparable DLs to standard LAMP for all 4 sets of primers. Compared to conventional PCR, its DLs were 10 times more sensitive for toxR, trh1, and trh2 and 100 folds greater for tdh. LAMP-XO also had equivalent DL to qPCR for rpoD but 10 times less sensitive for tdh, trh1, and trh2. This finding demonstrated that the LAMP-XO was as sensitive as standard LAMP and qPCR, but much more sensitive than PCR.

Comparative detection limit of LAMP-XO, standard LAMP, conventional PCR, and qPCR, assays using spiked shrimp samples

Prior to testing the analytical sensitivity of the assays, V. parahaemolyticus-negative shrimp samples were separately spiked with the reference strains at the concentrations of 104–100 CFU/2.5 g spiked shrimp. Amplification of V. parahaemolyticus gDNA was not observed by qPCR in a negative control (a non-spiked sample). The DL of LAMP-XO for rpoD, tdh, and trh2 was 100 CFU/2.5 g spiked shrimp, and for trh1 primers was 103 CFU/2.5 g spiked shrimp (Table 4, Fig. 2A, top). All of the colorimetric results were in accordance with the AGE results except for trh1 detection where AGE could detect down to 102 and 101 CFU/2.5 g spiked shrimp (Fig. 2A, bottom). However, no distinct positive (yellow) test results were observed at these dilutions. The detection limits (DLs) of standard LAMP (Fig. 2B) and qPCR assays (Fig. 2D) for all 4 sets of primers were 100 CFU/2.5 g spiked shrimp whereas those of PCR were 100 and 101 CFU/2.5 g spiked shrimp for toxR and tdh, respectively and 103 CFU/2.5 g spiked shrimp for trh1 and trh2 (Fig. 2C).

Table 4 Comparative detection limit of LAMP-XO, standard LAMP, conventional PCR, and qPCR assays using a 10-fold serial dilution of spiked shrimp samples (104–100 CFU/mL).

Primer	10-fold serial dilution of spiked samples (10 4 – 10 0 CFU/mL)	
	LAMP-XO	Standard LAMP	Conventional PCR	qPCR	
toxR/rpoD	100	100	100	100	
tdh	100	100	101	100	
trh1	103	100	103	100	
trh2	100	100	103	100	

Figure 2 Comparative sensitivity of LAMP-XO, standard LAMP, conventional PCR, and qPCR assays for the detection of total V. parahaemolyticus and pathogenic V. parahaemolyticus using 104–100 CFU/2.5 g spiked shrimp.

The rpoD primer was used for total V. parahaemolyticus detection in LAMP-XO, standard LAMP, and qPCR assays while the toxR primer was used in PCR assay. The tdh, trh1, and trh2 primers were used for pathogenic V. parahaemolyticus detection in all assays. (A) LAMP-XO assay detected by the naked eye (top) and AGE (bottom); (B) Standard LAMP assay detected by a turbidimeter (top) and AGE (bottom); (C) Conventional PCR detected by AGE; (D) Quantitative PCR (q-PCR) assay. L1: 104 CFU/2.5 g spiked shrimp, L2: 103 CFU/2.5 g spiked shrimp, L3: 102 CFU/2.5 g spiked shrimp, L4: 101 CFU/2.5 g spiked shrimp, L5: 100 CFU/2.5 g spiked shrimp, M: 1 kb DNA ladder, N: Negative control.

LAMP-XO assay revealed comparable DLs to standard LAMP and qPCR for rpoD, tdh, and trh2 and 1,000 times less sensitive than that of qPCR for trh1. Compared to conventional PCR, limits of LAMP-XO detection were 10 times more sensitive for tdh and 1,000 times more sensitive for trh2. These findings were in accordance with those using a 10-fold serial dilution gDNA which demonstrated overall LAMP-XO was more sensitive than PCR whereas LAMP-XO sensitivity was comparable to those of standard LAMP and qPCR except for trh1.

LAMP-XO assay for the detection of total V. parahaemolyticus (pathogenic and non-pathogenic strains) and pathogenic V. parahaemolyticus (tdh+ and/or trh1+ and/or trh2+) in raw seafood samples

LAMP-XO assay efficacy was validated using 102 raw seafood samples purchased from fresh markets and supermarkets and collected from shrimp farms in Thailand and North Vietnam. The colorimetric results were compared with the results from PCR and qPCR. Using qPCR as a gold standard, LAMP-XO and PCR assays identified 76/102 and 74/102 samples positive for V. parahaemolyticus contamination, respectively (Table 5). Thus, the clinical sensitivity, specificity, PPV, NPV, diagnostic accuracy, and percent overall agreement (POA) were 100% for all categories for LAMP-XO, but were 97.4%, 100%, 100%, 96.4%, 98.4%, and 98.0%, respectively, for PCR. No statistically significant difference was observed for V. parahaemolyticus detection between qPCR and LAMP-XO and between qPCR and PCR (P > 0.05).

Table 5 Comparison of qPCR, LAMP-XO, and conventional PCR for V. parahaemolyticus detection (n = 102 seafood samples).

	No. of results for
qPCR							
Method
and result	Positive	Negative	Sensitivity, %
(95% CI)	Specificity, %
(95% CI)	PPV, %
(95% CI)	NPV, %
(95% CI)	Diagnostic
accuracy (%)	POA (%)	
qPCR									
Positive	76	0							
Negative	0	26	100 (95.3–100)	100 (86.8–100)	100	100	100	100	
LAMP-XO									
Positive	76	0							
Negative	0	26	100 (95.3–100)	100 (86.8–100)	100	100	100	100	
PCR									
Positive	74	0							
Negative	2	26	97.4 (90.8–99.7)	100 (86.8–100)	100	96.4 (87.0–99.0)	98.4	98.0	

For pathogenic strain detection, 76 samples positive for V. parahaemolyticus by a gold standard qPCR were tested for tdh, trh1, and trh2 genes. LAMP-XO for tdh detection had sensitivity, specificity, PPV, and NPV of 90.5%, 100%, 100%, and 88%, respectively. The diagnostic accuracy and POA between LAMP and qPCR results were 94.4% and 97.4%, respectively (Table 6). PCR for tdh diagnosis showed 38.1% sensitivity, 98.2 specificity, 96.8% PPV, and 52.4% NPV. The diagnostic accuracy and POA between PCR and qPCR results were 62.7% and 81.6%, respectively (Table 6). No statistically significant difference was observed for tdh detection between qPCR and LAMP-XO (P > 0.05) while there were statistically significant differences between qPCR and PCR and between LAMP-XO and PCR (P < 0.05).

Table 6 Comparison of qPCR, LAMP-XO, and conventional PCR for tdh detection in 76 V. parahaemolyticus-positive samples.

	No. of results for
qPCR							
Method
and result	Positive	Negative	Sensitivity, %
(95% CI)	Specificity, %
(95% CI)	PPV, %
(95% CI)	NPV, %
(95% CI)	Diagnostic
accuracy (%)	POA (%)	
qPCR									
Positive	21	0							
Negative	0	55	100 (83.9–100)	100 (93.5–100)	100	100	100	100	
LAMP-XO									
Positive	19	0							
Negative	2	55	90.5 (69.6–98.8)	100 (93.5–100)	100	88 (66.1–96.5)	94.4	97.4	
PCR a									
Positive	8	1							
Negative	13	54	38.1 (18.1–61.6)	98.2 (90.3–100)	96.8 (80–99.7)	52.4 (44–60.7)	62.7	81.6	
Notes.

a There was a statistically significant difference for tdh detection between qPCR and PCR and between LAMP-XO and PCR (P < 0.05).

LAMP-XO for trh1 detection demonstrated 75% sensitivity, 100% specificity, 100% PPV, and 73.5% NPV. The diagnostic accuracy, and POA between LAMP-XO and qPCR results were 85.3% and 96.1%, respectively (Table 7). PCR for trh1 detection exhibited sensitivity, specificity, PPV, and NPV of 58.3%, 100%, 100%, and 62.5%, respectively. The diagnostic accuracy and POA between PCR and qPCR results were 75.4% and 93.4%, respectively (Table 7). No statistically significant difference was observed for trh1 detection between qPCR and LAMP-XO and between qPCR and PCR (P > 0.05).

Table 7 Comparison of qPCR, LAMP-XO, and conventional PCR for trh1 detection in 76 V. parahaemolyticus-positive samples.

	No. of results for
qPCR							
Method
and result	Positive	Negative	Sensitivity, %
(95% CI)	Specificity, %
(95% CI)	PPV, %
(95% C1I)	NPV, %
(95% CI)	Diagnostic
accuracy (%)	POA (%)	
qPCR									
Positive	12	0							
Negative	0	64	100 (73.5–100)	100 (94.4–100)	100	100	100	100	
LAMP-XO									
Positive	9	0							
Negative	3	64	75 (42.8–94.5)	100 (94.4–100)	100	73.5 (51.1–88.1)	85.3	96.1	
PCR									
Positive	7	0							
Negative	5	64	58.3 (27.7–84.8)	100 (94.4–100)	100	62.5 (46.1–76.5)	75.4	93.4	

LAMP-XO for trh2 detection exhibited 100% for sensitivity, specificity, PPV, NPV, diagnostic accuracy, and POA which were similar to those of qPCR. For trh2 detection, PCR revealed 55.6% sensitivity, 100% specificity, 100% PPV, and 61% NPV. The diagnostic accuracy and POA between PCR and qPCR results were 73.8% and 89.5%, respectively (Table 8). The statistically significant difference was observed for trh2 detection between qPCR and PCR and between LAMP-XO and PCR (P < 0.05).

Table 8 Comparison of qPCR, LAMP-XO, and conventional PCR for trh2 detection in 76 V. parahaemolyticus-positive samples.

	No. of results for
qPCR							
Method
and result	Positive	Negative	Sensitivity, %
(95% CI)	Specificity, %
(95% CI)	PPV, %
(95% CI)	NPV, %
(95% CI)	Diagnostic
accuracy (%)	POA (%)	
qPCR									
Positive	18	0							
Negative	0	58	100 (81.5–100)	100 (93.8–100)	100	100	100	100	
LAMP-XO									
Positive	18	0							
Negative	0	58	100 (81.5–100)	100 (93.8–100)	100	100	100	100	
PCR a									
Positive	10	0							
Negative	8	58	55.6 (30.8–78.5)	100 (93.3–100)	100	61 (48.3–72.4)	73.8	89.5	
Notes.

a There was a statistically significant difference for trh2 detection between qPCR and PCR and between LAMP-XO and PCR (P < 0.05).

Overall, LAMP-XO yielded results comparable to those of qPCR for rpoD, tdh, trh1, and trh2 detection since no statistically significant difference was observed between the 2 methods. Compared to PCR, LAMP-XO significantly demonstrated greater performance for tdh and trh2. For trh1, LAMP-XO was prone to have higher performance to PCR. However, no statistically significant difference was observed for trh1 detection between the 2 methods.

Distribution of pathogenic genes in V. parahaemolyticus-positive samples

Based on the results of a gold standard qPCR and the presence or absence of the tdh or trh1 or trh2 toxin genes, the 76 V. parahaemolyticus-positive samples could be classified into 8 groups: tdh+/trh1−/trh2−, tdh+/trh1+/trh2−, tdh+/trh1−/trh2+, tdh+/trh1+/trh2+, tdh−/trh1+/trh2−, tdh−/trh1+/trh2+, tdh−/trh1−/trh2+, and tdh−/trh1−/trh2− (Table 9). Most of V. parahaemolyticus isolates were tdh−/trh1−/trh2− (64.5%, 49/76) and the rest were pathogenic strains with tdh+ and/or trh1+ and/or trh2+. Among pathogenic strains (27 isolates), tdh+/trh1−/trh2+ strains were predominant (11 isolates) followed by tdh+/trh1+/trh2− (6 isolates), tdh+/trh1+/trh2+ (3 isolates), tdh−/trh1−/trh2+(3 isolates), tdh−/trh1+/trh2− (2 isolates), tdh+/trh1−/trh2− (1 isolate), and tdh−/trh1+/trh2+ (1 isolate).

Table 9 Characteristics of 76 V. parahaemolyticus isolates.

Genotype	No. of isolates (%)	
tdh + /trh1 − /trh2 −	1 (1.3%)	
tdh + /trh1 + /trh2 −	6 (7.9%)	
tdh + /trh1 − /trh2 +	11 (14.5%)	
tdh + /trh1 + /trh2 +	3 (3.95%)	
tdh − /trh1 + /trh2 −	2 (2.6%)	
tdh − /trh1 + /trh2 +	1 (1.3%)	
tdh − /trh1 − /trh2 +	3 (3.95%)	
tdh − /trh1 − /trh2 −	49 (64.5%)	

Discussion

Hygiene problems with seafood from cross-contamination in the seafood harvesting period from farm to fork lead to the increase of V. parahaemolyticus in the food chain (Hara-Kudo & Kumagai, 2014). To assure the safe seafood supply and to prevent economic losses, early monitoring and surveillance of V. parahaemolyticus are of utmost importance. In this study, a colorimetric assay based on LAMP-XO (Jaroenram, Cecere & Pompa, 2019) was developed, evaluated, and validated as an effective molecular tool to detect total and pathogenic V. parahaemolyticus in seafood. The factors, including the concentrations of dNTP mix, betaine, MgSO4, Bst 2.0 WarmStart DNA polymerase, and XO as well as reaction temperature and reaction time were optimized to obtain maximal amplification for each target gene. To accelerate the reaction amplification, we attempted to design the extra primers called multiple hybrid, inner primers (MHP) for rpoD, tdh, trh1, and trh2 target genes. However, only MHP for rpoD were accomplished. The novel molecular method called “MHP-LAMP” developed by our group could be used successfully to increase the sensitivity and speed up the rpoD gDNA amplification (Lamalee et al., 2023). The MHP for the other 3 target genes could not be designed due to the limitation of the sequence length and sequence properties of the core primers.

Using a 10-fold serial dilution of gDNA, the DL of LAMP-XO for rpoD and tdh detection was 102 copies/reaction and for trh1 and trh2, it was 103 copies/reaction. In sterilized Pacific white shrimp spiked with known quantities of the reference strains, LAMP-XO detected 100 CFU/2.5 g spiked shrimp for rpoD, tdh, and trh2 and 103 CFU/2.5 g spiked shrimp for trh1 within 4 h of pre-enrichment. Although no distinct positive (yellow) test results of LAMP-XO were observed for trh1 at 102 and 101 CFU/2.5 g spiked shrimp, LAMP-XO products were observed at these dilutions by AGE. The discrepancy between LAMP-XO and AGE results was due to the fact that XO in LAMP reaction could not shift from purple to yellow because pH of LAMP reaction was > 6.7 due to the low amount of LAMP reaction by-products. These imply that the threshold of LAMP amplicons to trigger a distinctive purple-to-yellow readout is higher than that to allow a clearly visible result on AGE. Extending an incubation time to more than 90 min may help by promoting a color result development. Unlike rpoD, tdh, and trh2 primers, the trh1 primers lack of the LB primer to accelerate the reaction amplification resulting the decrease of the sensitivity of the trh1-LAMP-XO assay. When compared to conventional PCR, standard LAMP, and qPCR as a gold standard method, our LAMP-XO yielded results comparable to standard LAMP and qPCR except for trh1 in spiked samples. However, the DL of LAMP-XO for trh1 in spiked samples could be improved by increasing the reaction time to 90 min to increase LAMP reaction amplification. Compared to conventional PCR, LAMP-XO had greater sensitivity for tdh and trh2 detection.

For the detection of V. parahaemolyticus using a 10-fold serial dilution of gDNA, our LAMP-XO demonstrated similar sensitivity to the study by Hu et al. (2021) (1.127 × 102 copies/reaction) and demonstrated higher sensitivity than that of the study by Liu et al. (2017) (1.789 × 103 copies/reaction). For the detection of tdh using a 10-fold serial dilution of gDNA, the LAMP-XO result had comparable sensitivity to that of the study by Anupama et al. (2021) (1.82 × 102 copies/reaction). In spiked samples, the LAMP-XO result for V. parahaemolyticus detection showed higher sensitivity than those of the previous studies by Di et al. (2015) (2 CFU/g of 3-h spiked sample) and Zeng et al. (2014) (1.9 CFU/g of 6-h spiked sample).

To confirm the clinical sensitivity, specificity, and accuracy of LAMP-XO assay, 102 raw seafood samples were used. The results revealed that the performance of LAMP-XO assay for V. parahaemolyticus rpoD, tdh, trh1, and trh2 detection was comparable to a gold standard qPCR and significantly superior to conventional PCR for tdh and trh2 detection as analyzed by a McNemar chi-square test. These results indicate that the LAMP-XO assay developed in this present study is a better choice than PCR and qPCR for routine detection of V. parahaemolyticus in naturally contaminated seafood samples and in environment since this method does not require expensive equipment and well-trained personnel and has a short turnaround time.

The occurrence of tdh and/or trh in environmental V. parahaemolyticus isolates is normally 1–10% depending on locations, sample sources, and detection methods (Raghunath, 2015). The LAMP-XO results showed that 35.5% (27/76) of V. parahaemolyticus detected in this present study were pathogenic strains (tdh+ and/or trh1+ and/or trh2+). The tdh+/trh1−/trh2+ strain was the most frequently observed (11/76;14.5%). In this finding, the coexistence of trh1 and trh2 were observed in tdh+/trh1 +/trh2+ (3/76; 3.95%) and tdh−/trh1+/trh2+ (1/76; 1.3%) strains. The tdh+/trh1+/trh2+ and tdh−/trh1+/trh2+ strains were obtained from fresh market and supermarket samples, respectively. However, previous studies demonstrated that V. parahaemolyticus isolates carrying trh2 did not contain trh1 (Kishishita et al., 1992; Kongrueng et al., 2018). The discrepancy of the results is probably due to mixed populations of V. parahaemolyticus carrying trh1 and V. parahaemolyticus carrying trh2 in the same samples since the detection of rpoD, tdh, trh1, and trh2 in these 4 samples was done directly from the seafood samples after enrichment. In this present study, the number of trh+ V. parahaemolyticus was prone to be higher than that of tdh+ V. parahaemolyticus.

It is worth noting that although we have reported the LAMP primer set (Lamalee et al., 2023), this does not lower the value of the present study. This is because our previous publication focuses on introducing the new concept of using additional hybrid LAMP primers to enhance the diagnostic sensitivity of a typical LAMP assay, and validate it at a proof-of-concept level by using rpoD-LAMP for V. parahaemolyticus detection as a fundamental model. However, in our present study, we extended our finding by transforming it into a colorimetric, xylenol orange (XO)-based LAMP assay with the naked-eye readout format to enable simplicity yet having detection efficiency as a more complicated PCR-based protocol. To the best of our knowledge, LAMP-XO has not been applied to detect V. parahaemolyticus. This brings about the novelty of this study to some extent. In addition, the significance of this study is that the LAMP-XO assay showed 100% reliability in detecting both pathogenic and non-pathogenic V. parahaemolyticus (by rpoD and trh2 genes) as well as discriminating them (by trh2 gene). Thus, they could bridge the gap by complementing or replacing the current diagnostic methods as a quick and reliable assay while confirmatory V. parahaemolyticus diagnosis are processed by slower and more expensive conventional methods such as real-time PCR.

Conclusions

Global outbreaks caused by pathogenic V. parahaemolyticus are recurrent, emphasizing the requirement for effective control of contaminants in seafood. The LAMP-XO assay reported here is rapid, simple, practical, cost-effective, and as efficient as qPCR. Thus, this assay is suitable to facilitate surveillance for total V. parahaemolyticus (pathogenic and non-pathogenic strains) and pathogenic V. parahaemolyticus (tdh+ and/or trh1+ and/or trh2+) contamination in seafood, screening of contaminated seafood prior to consumption, and examinations to detect the food poisoning causative agents. This assay is also useful for ecological research related to environmental factors, seasons, areas, and practices. From our experience, this assay can be further improved to make it more efficient for the weak positive reaction, for example, by increasing the amplification time for trh1 from 75 min to 90 min and increasing the XO concentration for trh2 from 0.03 mM to 0.06 mM. The use of a one-step LAMP-XO colorimetric assay together with the addition of the MHP based on the previously reported core primer set are the concepts that can be applied to boost sensitivity and rapidity of other existing LAMP-based assays. It could aid in reducing the cost and time in redesigning a whole new primer set from the beginning.

Supplemental Information

Supplemental Information 1 Reagent concentrations of the initial standard protocol for LAMP-XO optimization

Click here for additional data file.

Supplemental Information 2 Design of parameter optimization in LAMP-XO assay

Click here for additional data file.

Supplemental Information 3 PCR primers and conditions used in this study

Click here for additional data file.

Supplemental Information 4 QPCR primers and conditions used in this study

Click here for additional data file.

Supplemental Information 5 Comparison of the results of 102 raw seafood samples among LAMP-XO, conventional PCR, and qPCR assays

Click here for additional data file.

Supplemental Information 6 Comparison of LAMP-XO, conventional PCR and qPCR assays

Click here for additional data file.

Supplemental Information 7 Detection of total V. parahaemolyticus (pathogenic and non-pathogenic strains) and pathogenic V. parahaemolyticus (tdh+ and/or trh1+ and/or trh2+) in 102 raw seafood samples using LAMP-XO, PCR, a

Click here for additional data file.

We would like to thank the Institute of Food Research and Product Development, Kasetsart University, Bangkok, Thailand, and the Department of Microbiology, Faculty of Science, Mahidol University, Bangkok, Thailand, for supplying all reagents and equipment. We would also like to acknowledge Dr. Jennifer Elliman for reviewing and editing the manuscript.

Additional Information and Declarations

Competing Interests

Author Contributions

Data Availability

The authors declare there are no competing interests.

Aekarin Lamalee performed the experiments, analyzed the data, prepared figures and/or tables, and approved the final draft.

Soithong Saiyudthong conceived and designed the experiments, performed the experiments, authored or reviewed drafts of the article, and approved the final draft.

Chartchai Changsen performed the experiments, prepared figures and/or tables, and approved the final draft.

Wansika Kiatpathomchai conceived and designed the experiments, authored or reviewed drafts of the article, and approved the final draft.

Jitra Limthongkul performed the experiments, prepared figures and/or tables, and approved the final draft.

Chanita Naparswad performed the experiments, prepared figures and/or tables, sample collection, and approved the final draft.

Charanyarut Sukphattanaudomchoke performed the experiments, prepared figures and/or tables, sample collection, and approved the final draft.

Jarinya Chaopreecha performed the experiments, prepared figures and/or tables, sample collection, and approved the final draft.

Saengchan Senapin performed the experiments, authored or reviewed drafts of the article, sample collection, and approved the final draft.

Wansadaj Jaroenram conceived and designed the experiments, analyzed the data, prepared figures and/or tables, authored or reviewed drafts of the article, and approved the final draft.

Sureemas Buates conceived and designed the experiments, performed the experiments, analyzed the data, prepared figures and/or tables, authored or reviewed drafts of the article, and approved the final draft.

The following information was supplied regarding data availability:

The raw data are available in the Supplementary File.

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
