# Peer review of "End-point rapid detection of total and pathogenic Vibrio parahaemolyticus (tdh+ and/or trh1+ and/or trh2+) in raw seafood using a colorimetric loop-mediated isothermal amplification-xylenol orange technique"

_PeerJ, doi:10.7717/peerj.16422_

## Round 0.1 · original submission · Major Revisions

Dear Authors

There have been a number of questions raised by the reviewers. Could you please address those questions.

Reviewer 1 ·

Basic reporting

The authors developed a colorimetric LAMP-XO technique aiming to detect pathogenic and non-pathogenic Vibrio parahaemolyticus in raw seafood. They claimed that LAMP-XO targeted rpoD for species specificity, and tdh, trh1, and trh2 for pathogenic strains.
Vibrio parahaemolyticus stains contained many virulence factor genes such as tdh, trh, tlh, pir,or toxR. Three kinds of haemolysin TDH, TRH and TLH, encoded by tdh, trh and tlh genes, responsible for haemolytic activity are usually present in clinical V.parahaemolyticus strains causing human diseases. The absence of tdh and trh genes could not exclude the probability of haemolysis or the pathogenicity when invading hosts. Secretion systems are of great significance for virulence by directing toxins to the bacterial surface or host cells. Vibrio spp. also secreted proteases, such as serine protease, collagenase, chitinase, lipase and peptidase, probably involved in destruction of the host barrier. Based on the absence of the tdh and trh, it is inaccurate to identify non-pathogenic V. parahaemolyticus strains. In this paper, the authors just identified the absence or presence of tdh and trh genes in V.parahaemolyticus strains using the LAMP-XO method. In addition, the paper did not provide any results for the specificity of LAMP-XO primer sets. The manuscript should be re-organized and improve the clarity. I also have additional concerns as follows:
Line 163, The abbreviation “TSB” is first used, please give the full name.
Line 167, Change subject to “subjected”.
Lines 176- 180, The TSB was used to isolate the bacteria, how did know the isolates were V. parahaemolyticus strains? How did you identify the V. parahaemolyticus strains?
Lines 184-185, what about the specificity for LAMP-XO, standard LAMP, qPCR and conventional PCR?
Line 185-189, “For confirmation”, I am confused what you confirmed? Only based on the colonies grown on the TCBSA and CHROMagarTM Vibrio mediums, how can you confirm?
Line 223, “gDNA of 106 copies/μL/reaction”, how did you determine the “DNA copy”? Likewise, Lines 225, 271.
Lines 243, 256, what about the reaction program of conventional PCR and quantitative PCR (q-PCR)?
Line 280, “inoculating levels of 104-100 CFU/2.5 g sample”, how did you determine the inoculating levels?
Lines 289-295, Why was the qPCR used as a gold standard method? How did you define the “clinical sensitivity” , “specificity”, “positive predictive values”, “negative predictive values”, “diagnostic accuracy”, and “percent overall agreement” ?
Lines 303-304, how did you optimize LAMP reagents, reaction times, and XO concentrations? what about the result? Can you provide some images to show the results?
Lines 362-364, “Using qPCR as a gold standard ……...positive for V. parahaemolyticus contamination, respectively”. What are the qPCR detection results?
Lines 364-366, “the sensitivity, specificity…….... and 98.0%, respectively, for PCR.” “Sensitivity” and “specificity”” are very confusing. I cannot understand the statement “the sensitivity was 100%”. Likewise, for “specificity”. How to define the sensitivity or specificity? And elsewhere, Lines 371-375, 381-383, 388-389.
Line 368, No statistically significant ……... between qPCR and PCR (P > 0.05)”. What was compared for the statistically significant difference? And elsewhere, Lines 375-376, 384, 390.
Line 398, What about the LAMP-XO result for the distribution of pathogenic genes in Vibrio parahaemolyticus. Was the LAMP-XO result comparable to the qPCR results?
Line 434, The sentence “more than 90 min may help promoting a color result development”, is this result experimentally obtained? Or is it supported by references? Please make it clear.
In the “discussion” section, there are many repetitive descriptions with the “results” section.
Lines 472-485, The conclusion should be drawn based on the experiments and results obtained in the paper. I suggest rewriting the conclusion.

References:
Ghenem L, Elhadi N, Alzahrani F, et al. Vibrio parahaemolyticus: a review on distribution, pathogenesis, virulence determinants and epidemiology. Saudi J Med Med Sci 2017; 5:93–103.
Wang R, Zhong Y, Gu X, et al. The pathogenesis, detection, and prevention of Vibrio parahaemolyticus. Frontiers in Microbiology, 2015, 6:144.
Zhang X, Sun J, Chen F, et al., Phenotypic and genomic characterization of a Vibrio parahaemolyticus strain causing disease in Penaeus vannamei provides insights into its niche adaptation and pathogenic mechanism. Microbial Genomics, 2021;7:000549

Experimental design

no comment

Validity of the findings

'no comment

Reviewer 2 ·

Basic reporting

no comment

Experimental design

no comment

Validity of the findings

The authors have standardized a LAMP assay using a pH-sensitive dye, xylenol orange, which can detect the pathogenic and non-pathogenic strains of Vibrio parahaemolyticus in raw seafood.

Title: avoid abbreviation in the title. LAMP-XO can be expanded
Line 475, in the title and other places in the manuscript: ……rapid, simple, practical, cost-effective, and as efficient as the gold standard qPCR. The study does not compare the rapidity and cost-effectiveness of LAMP-XO with other methods as such. Hence, the statement can be avoided or modified accordingly.
Line 157: expand gDNA for the first time
Line 158: The study does not contain any other vibrio culture as a negative control.
Line 161: Was the Vp10/5 (tdh+/trh1+) isolated in this study? If not, the source can be mentioned.
Line 173: Of the 102 samples, 86 were…….which raw seafood samples are these?
Line 175 and 176: Samples were collected in 2018, and 2013 respectively. How were these samples stored/maintained? Storage information can be mentioned as these are not fresh samples.
Lne18-18: For confirmation of the presence of V. parahaemolyticus, TCBSA followed by CHROMagar Vibrio was used. Why not confirm by a simple molecular method like V. parahaemolyticus-specific PCR?
Line 269: mention the name of the strain
Line 276: lack of bacterial contamination was confirmed by the culture method using TCBS. Why not use molecular methods like standard LAMP, conventional PCR, and/or qPCR? Molecular method can still identify the presence of bacteria, even if it is negative in the culture method. Spiked shrimp sample results data (table 4) showed better sensitivity than Table 3.
Line 285: Differentiation of pathogenic and non-pathogenic V. parahaemolyticus in raw seafood samples by LAMP-XO. This heading is misleading. The assay does not differentiate, it just detects the pathogenic and non-pathogenic Vibrio parahaemolyticus as multiple assay set-ups were carried out using both pathogenic and non-pathogenic specific primers separately. Also, modify accordingly in lines 151-152 and other places in the manuscript.
Line 322: rpoD and tdh primers indicated positive amplification at 104-102 copies, while trh1 and trh2 primers did at 104-103 copies. Why is a range mentioned?
Table 2 information is available in materials and methods. The table can be provided as a supplementary table.
Figures 1 and 2: (A) trh1 gene: LAMP-XO assay detected by the naked eye is unclear. The colour change is not visible.

---

## Round 0.2 · Major Revisions

A major concern is the integration of the response to reviewer 1 comments not being addressed in the revised manuscript. Could you please address this and the concerns of reviewer 2?

Reviewer 1 ·

Basic reporting

In most cases, the authors responded to the comments, but barely integrated the responses into the manuscript. Most defects have not been addressed in the revision.

The term “total and pathogenic Vibrio parahaemolyticus” mentioned in the revision is confusing.
The LAMP-XO primer sets used in this study were previously reported; this lowered the novelty of the study. I could not find the reference “A. Lamalee (a manuscript in revision, 2023)” you mentioned.
The CHROMagarTM medium cannot accurately identify the V. parahaemolyticus strains.
The definitions for the terms “clinical sensitivity” and “specificity” were confusing.

Experimental design

The LAMP-XO primer sets used in this study were previously reported; this lowered the novelty of the study.

Validity of the findings

no comment

Additional comments

no comment

Reviewer 2 ·

Basic reporting

Not applicable

Experimental design

The authors have addressed my comments; however, a few points still require clarification. My responses to the author's response are listed below.

1. As suggested by the reviewer, the title has been changed to “End-point rapid detection of total and pathogenic Vibrio parahaemolyticus (tdh+ and/or trh1+ and/or trh2+) in raw seafood using a colorimetric loop-mediated isothermal amplification (LAMP)-xylenol orange (XO) technique”.

Response: Abbreviations in brackets, i.e. (LAMP) and (XO), can be removed.


2. The rapidity and cost-effectiveness of LAMP have been widely compared with qPCR and conventional PCR methods as previously described (Kokkinos et al., 2014; Oriero et al., 2015; Hu et al., 2020). The rapidity and cost-effectiveness are the advantages of LAMP and are among the reasons of using LAMP technique in this present study.

Response: It is understood that LAMP assays have been widely compared with qPCR and conventional PCR methods in various other studies and found to be rapid and cost-effective. But your study (the present study) does not compare the time required for the study and cost requirement analysis comparison. Hence, based on previously published LAMP work, you cannot conclude your LAMP-XO assay is rapid and cost-effective. It can be discussed in general in the manuscript, but not in the conclusion as a statement unless you have performed this in your own study or it is a review article.
The word “gold standard” for qPCR can be removed.


4. Line 160: The specificity of the primers used in this present study has been already tested with other Vibrio and other food-borne pathogens as described in Comment 7 of Reviewer 1. Therefore, only DW was used as a negative control to check the contamination of the reaction in this present study.
Response: What is DW?


8. Line 196: The words “For confirmation” has been changed to “For V. parahaemolyticus isolation”.

Response: In the manuscript, line 195: it is still “For further confirmation.”


10. Lines 295-296: In the spiking experiment, the culture method was used to confirmed V. parahaemolyticus contamination since we would like to detect the viable and culturable V. parahaemolyticus not the V. parahaemolyticus gDNA.

Response: You are performing a nucleic acid-based amplification assay. The presence of V. parahaemolyticus genomic DNA can mislead the assay sensitivity.

Validity of the findings

Not applicable

Additional comments

Not applicable

---

## Round 0.3 · accepted · Accept

Dear Authors

I have reviewed your responses to the reviews and you have addressed all comments and your manuscript is now ready for publication.

Regards
Travis